# A consensus-based decision model for assessing the health systems

**Yang Xu[1], Kin Keung Lai[2]*, W. K. J. Leung[3]**

**1** School of Economics & Management, Xi'an Technology University, Xi'an, China, **2** College of Economics, Shenzhen University, Shenzhen, China, **3** Department of Marketing, City University of Hong Kong, HKSAR, Kowloon Tong, China

* mskklai@outlook.com

## Abstract

Many countries and international organisations have been developing health system performance assessment frameworks and indicators to support healthcare management and inform public health policy. Effectiveness, accessibility, safety and patient-centeredness were four dimensions that were most commonly measured. This paper develops a new consensus-based decision model to assess the health systems, in which different stakeholders of healthcare systems are identified by different decision approaches, i.e., the coefficient variation approach, the Shannon entropy approach and the distance-based approach, respectively. The consensus result is obtained by minimizing the total deviation from the ideal point. A numerical illustration with simulated data is presented to show the effectiveness of our model.

## 1. Introduction

Around the world, health systems play a central role in helping human beings maintain and improve their health conditions. Nowadays, the emphases on health system reform and achievement have resulted in an increased awareness about the significance of strengthening health systems and the importance of assessing the health systems [1, 2]. By assessing the performance of health systems, policy-makers could have a better understanding of how the health systems work, and therefore suggest actions to improve quality of the services for the health of population. Performance assessment not only offers the transparency for securing accountability for health systems, but also identifies the weaknesses of the functioning of health systems for improvement. Some common objectives of an assessment include, but not limited to, identifying good and bad health practice, enhancing effectiveness and accessibility of care services, and improving the safety of patients.

Indicators developed by international organisations could similarly be categorised into different aspects of a health system for evaluation. WHO 100 Core Health Indicators could be grouped into four domains, including health status, risk factors, service coverage and health system [3], whereas 88 ECHI are grouped under the following headings: (i) demographic and socio-economic situation, (ii) health status, (iii) health determinants, (iv) health interventions: health services, and (v) health interventions: health promotion [4]. Three of the four tiers of

**Funding:** This research is financially supported by Humanities and Social Sciences Fund of the Ministry of Education (No. 18YJAZH080), and

Project of Science and Technology Department of Shaanxi Province (No. 2020JQ-654).

**Competing interests:** The authors have declared that no competing interests exist.

the OECD Health Care Quality Framework are identical to those domains in the Australian framework, while the major difference between these two frameworks is the inclusion of a component of health system design, policy and context for evaluation in the OECD Framework [5].

There exist a large number of studies on health system assessment. Schieber et al. [6] and Anderson and Hussey [7] present data from Organization for Economic Co-operation and Development (OECD) and World Health Organization (WHO) on the performance of health systems in 29 industrialized countries, and also conduct the cross-national performance comparisons. A number of developed countries have initiated performance measurement for managing the output of healthcare services and monitoring the progress for achieving the goals of their health systems [8, 9]. Some international organisations, such as the WHO and OECD also take a lead in designing, advocating, and implementing health system performance measurement [10–12]. Schang et al. [13] employ ranking intervals and dominance relations to handle incomplete information about a set of weights, and therefore to develop robust composite measures of health quality. Roy et al. [14] develop a rough strength relational DEMATEL model for analyzing the key success factors of hospital service quality. Wang and Fu [15], Fu et al. [16], and Shen et al. [17] improve the Value Measure of health systems that is published by Royal Philips, by means of applying social choice theory and then proposing the Best-Worst method, Hurwicz criterion approach in conjunction with CRITIC method, and Stochastic Multicriteria Acceptability Analysis for group decision making (SMAA-2), respectively. Wong et al. [18] provide a comprehensive review about the national and international frameworks and indicators about health system performance assessment, and identified the effectiveness, accessibility, safety and patient-centeredness as the four components for evaluation. Kruk and Freedman [1] provide a comprehensive review of methods to assess health system performance in developing countries.

Dimensions of health care performance under the national and international frameworks are extracted from the framework covered most areas of health care quality suggested by the WHO [19] and Institute of Medicine [20]. Effectiveness, accessibility, safety, and responsiveness/patient-centeredness are four dimensions that are consistently monitored in health system performance [18]. Effectiveness refers to the degree of achieving the desired outcome, following the provision of health care services, while accessibility is the ease to reach a particular health service. Safety is a dimension that focusing the delivery of health care, which minimize risks and harm to the service users. Responsiveness/Patient-centeredness concerns whether the healthcare services take the preferences and aspirations of individual service users into account.

The present paper is motivated by the observation that in the process of performance assessment, not only the preferences associate with evaluation criteria may exhibit a substantial degree of variability, but also different members of the decision committee have different opinions, which are extremely difficult to achieve a group consensus [21, 22]. In this sense, this work proposes a consensus-based decision model to assess the health systems. Specifically, two issues should be addressed: (1) how to define the individual stakeholders of health systems? (2) how to achieve the consensus among different stakeholders? In this study, different stakeholders of health systems are identified by objective weight determination approaches, namely, the coefficient variation approach, the Shannon entropy approach and the distance-based approach. The main advantage of these objective approaches is the reduction of decision bias in terms of ignoring the subjective judgments of the individual stakeholders. Objective criteria weight determination approaches are usually applicable when individual stakeholders disagree on the exact values of criteria weights [23]. Specifically, the rationale behind objective criteria weight determination approaches is that the importance degree of a criterion is a function of

the information conveyed by this criterion, relative to a whole set of alternatives. To reduce the discrepancies among different stakeholders, we further develop a consensus-based model based upon the rationale minimizing the total deviation from the ideal point. Using different approaches to represent decision makers is not new and widely found in the decision literature [23], which is reasonably applied in the field of health system assessment, due to the fact that different assessment agencies definitely have different evaluation rationales. The model is developed because of the need for a rigorous, comprehensive health system assessment tool that is capable of connecting multiple components of health system to national health system performance indicators and national policy.

The main contribution of this study is proposing a novel consensus-based decision model to assess the health systems, in terms of addressing the aforementioned two research issues. In comparison with the existing models for assessing the health systems, the proposed consensus-based decision model has three distinct characteristics. First, the individual stakeholders' assessments are made based on the same set of criteria to formulate a multiple criteria group decision making (MCGDM) framework. This makes the decision results more objective than conventional single-person decision. Second, the criteria weights are determined based solely on the dataset itself, which can effectively reduce the decision bias and to some extent improve the decision quality. Third, the consensus-reaching method is easy-to-understand and simple-to-implement.

The rest of this paper proceeds as follows. Section 2 develops the aforementioned consensus-based decision model. Section 3 provides a numerical illustration. Section 4 concludes this study and provides some future research directions.

## 2. The consensus-based decision model

The framework developed to assess the performance of health systems in terms of multiple key components is depicted in Table 1 below, in which the normalized input element $x_{ij}, i = 1,2,\ldots, m, j = 1,2,\ldots, n$ denoting the performance of health system $i$ in terms of component $j$. All input data $x_{ij}$ have been normalized from the raw data $y_{ij}$ into 0–1 scale using $x_{ij} = \frac{y_{ij}}{\sum_{i=1}^{m} y_{ij}}$. Meanwhile, all components are assumed to be benefit-type, while the cost-type components could be take the reciprocal or negativity transformation.

In a general form, the overall performance of a typical health system can be obtained using a simple additive weighted value function, which is known as the underlying model for most Multiple Criteria/Attribute Decision Making methods, as below:

$$S_i = \sum_{j=1}^{n} x_{ij} w_j.$$

**Table 1. The assessment framework.**

| Health systems | Components | | | |
|:---:|:---:|:---:|:---:|:---:|
| | **1** | **2** | ... | **n** |
| 1 | $x_{11}$ | $x_{12}$ | ... | $x_{1n}$ |
| 2 | $x_{21}$ | $x_{22}$ | ... | $x_{2n}$ |
| ... | ... | ... | ... | ... |
| m | $x_{m1}$ | $x_{m2}$ | ... | $x_{mn}$ |

The assessment of health systems are not limited to the requirements and concerns of heath service providers, whose primary concerns represent the assurance of their own economic well-being and ability to proactively operate as well as the development of sustainable strategies to realize their own interests. In addition, another stakeholders of health systems with competing objective of health service providers, i.e., financiers, should be incorporated when systematically evaluating health systems. Besides health service providers and financiers striving for the realization of their concerns, patients also have an important say in this potential conflict and wish to incorporate their interests [24].

The general framework for the proposed consensus-based decision model is presented as below:

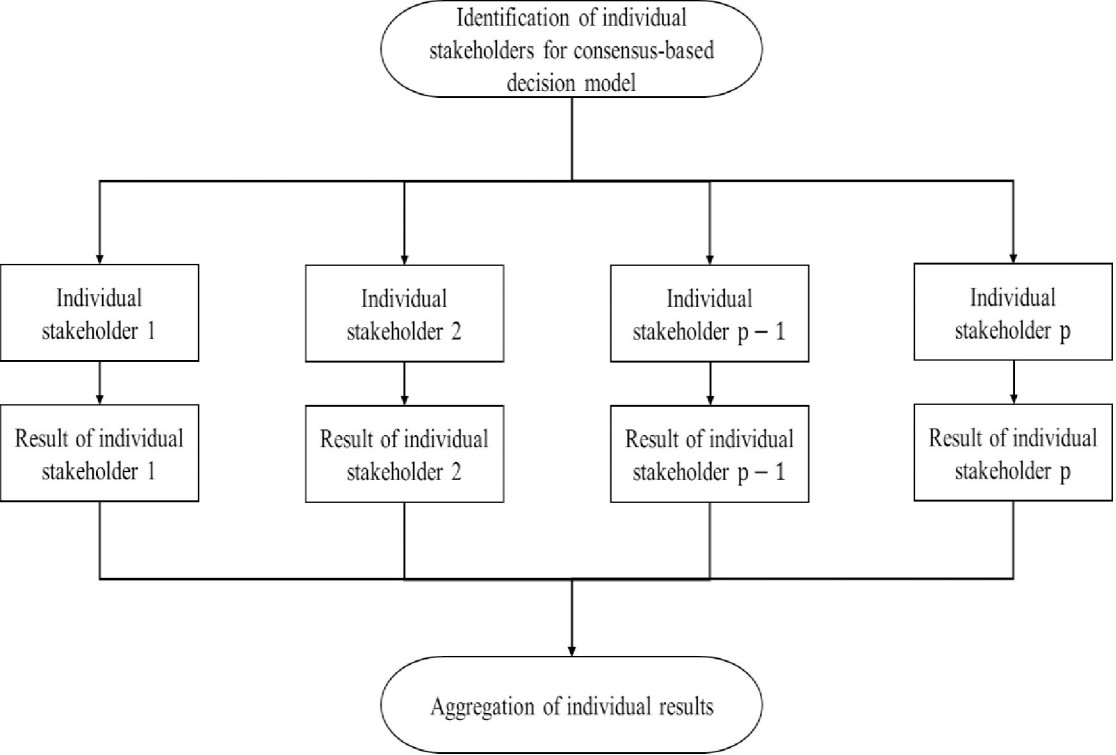

**Fig 1. The general framework of consensus-based decision model.**

According to Fig 1, the propose consensus-based decision model consists of two main phases: identification of individual stakeholders and implementation of standard criteria weight determination process for each individual stakeholder, and aggregation of individual opinions.

In what follows, we obtain the results from different stakeholders of healthcare systems, while individual stakeholder is identified by a typical decision making approach to generating weights associated with the key components, then aggregating the performance of health systems.

## 2.1 The Shannon entropy approach

In information theory, Shannon entropy is the expected value of the information contained in each message [25]. In the field of decision making, Shannon entropy has been proved to be a useful and effective mathematical concept to determine weights [23, 26].

The Shannon entropy approach for component weight determination proceeds in the following three steps:

i. Entropy calculation for $j$th component. The entropy value with respect to each component is represented as

$$e_j = -k \sum_{i=1}^{m} x_{ij} \ln(x_{ij}),\qquad(1)$$

where $k = \frac{1}{\ln(m)}$ is a constant.

ii. Dispersion computation for $j$th component. The measure of dispersion of the $j$th component is denoted by

$$\varphi_j = 1 - e_j.\qquad(2)$$

iii. Weight determination for $j$th component. On the basis of the dispersion measurement for $j$th component, the weights can be determined by

$$w_j^e = \frac{\varphi_j}{\sum_{j=1}^{n} \varphi_j}.\qquad(3)$$

## 2.2 The coefficient of variation approach

In probability theory and statistics, the coefficient of variation, alternatively known as relative standard deviation, is a standardized measure of dispersion of a probability distribution or frequency distribution, which has been widely applied in the areas of engineering and physics. The application of coefficient of variation approach to determine the weights is pioneered by Zeleny [27] and further developed by Pomerol and Barba-Romero [28] in the field of Multiple Criteria Decision Making.

The working process of coefficient of variation approach is demonstrated as follows:

i. i. Mean calculation. The mean value of regarding to $j$th component can be computed by

$$\bar{x}_j = \frac{\sum_{i=1}^{m} x_{ij}}{m}.\qquad(4)$$

i. ii. Standard deviation calculation. Using the input data and the mean values in (4), the standard deviation is calculated by

$$\sigma_j = \sqrt{\frac{1}{m} \sum_{i=1}^{m} (x_{ij} - \bar{x}_j)^2}.\qquad(5)$$

i. iii. Coefficient of variation calculation. In line with the definition of the coefficient of variation, that is defined as the ratio of the standard deviation ($\sigma_j$) to the mean ($\bar{x}_j$), the coefficient of variation associated with the $j$th component is computed for dispersion measurement:

$$\delta_j = \frac{\sigma_j}{\bar{x}_j}. \tag{6}$$

i. iv. Weight determination for $j$th component. Based upon the dispersion measurement obtained by (6), we compute the weight as

$$w_j^{cv} = \frac{\delta_j}{\sum\limits_{j=1}^{n} \delta_j}. \tag{7}$$

## 2.3 Distance-based approach

The distance-based approach proposed in this study is an extension of TOPSIS method [29], which considers the geometric distances compared with optimistic and pessimistic values, respectively. The distance-based approach process is carried out as follows:

i. Optimistic and pessimistic values determination with respect to $j$th component. The optimistic and pessimistic values for all components are defined as
optimistic values: $U^+ = (\max_i \{x_{i1}\}, \max_i \{x_{i2}\}, \ldots, \max_i \{x_{in}\})$,
pessimistic values: $U^- = (\min_i \{x_{i1}\}, \min_i \{x_{i2}\}, \ldots, \min_i \{x_{in}\})$.

ii. Distances computation. The geometric distance between the input data and the optimistic/pessimistic values for component $j$ are computed as

$$d_j^+ = \sqrt{\sum_{i=1}^{m} (x_{ij} - \max_i \{x_{ij}\})^2}, \tag{8}$$

$$d_j^- = \sqrt{\sum_{i=1}^{m} (x_{ij} - \min_i \{x_{ij}\})^2}. \tag{9}$$

iii. Dispersion measurement computation. On the strength of the above distances in (8) and (9), the dispersion measurements associated with all components are denoted by

$$\psi_j = \frac{d_j^+}{d_j^+ + d_j^-}, \tag{10}$$

the larger value of $\psi_j$, the more important the component $j$ is.

iv. Weight determination for $j$th component. Analogously, the weights associated with different components can be determined according to the measurements of dispersion, that is

$$w_j^d = \frac{\psi_j}{\sum\limits_{j=1}^{n} \psi_j}. \tag{11}$$

## 2.4 A consensus-based model

Recall that the opinions from stakeholders of healthcare systems are represented by different weight elicitation schemes proposed in subsections 2.1–2.4, the purpose of this subsection is to

develop a model for aggregating individual opinions into a group-level decision, in terms of minimizing the total deviation from the ideal point to reach a group consensus. The intuitively appealing principle behind minimizing the total deviation from the ideal point is that every health system definitely seeks to make the results determined by all decision makers as close to the ideal point as possible.

Motivated by Ma et al. [30], we formulate a weighted decision matrix $\Omega = [H_{il}]_{mL}$, where

$$H_{il} = z_{il}\lambda_l, i = 1, 2, \ldots, m, l = 1, 2, \ldots, L, \tag{12}$$

and $z_{il}$ shows the normalized result made by the decision maker $l$.

The ideal points are defined as $\Psi^* = \{\Psi_1^*, \Psi_2^*, \ldots, \Psi_L^*\}$, in which

$$
\begin{aligned}
\Psi_j^* &= \max\{\Psi_{1l}, \Psi_{2l}, \ldots, \Psi_{ml}\} \\
&= \max\{z_{1l}\lambda_l, z_{2l}\lambda_l, \ldots, z_{ml}\lambda_l\} \\
&= \max\{z_{1l}, z_{2l}, \ldots, z_{1l}\}\lambda_l \\
&= z_l^*\lambda_l,
\end{aligned}
\tag{13}
$$

and $z_l^*$ is the ideal value by decision maker $l$.

Considering each health system, the performance distance between individual decision maker's opinion and the ideal value is defined as follows:

$$
\begin{aligned}
D_i &= \sum_{l=1}^{L}(H_{il} - \Psi_l^*)^2 \\
&= \sum_{l=1}^{L}(z_{il} - z_l^*)^2 \lambda_l^2.
\end{aligned}
\tag{14}
$$

Therefore, a multi-objective programming is proposed to optimize the overall performance of all health systems:

$$
\begin{cases}
f_1 = \min \sum_{l=1}^{L}(z_{1l} - z_l^*)^2 \lambda_l^2 \\
f_2 = \min \sum_{l=1}^{L}(z_{2l} - z_l^*)^2 \lambda_l^2 \\
\qquad \cdots \\
f_m = \min \sum_{l=1}^{L}(z_{ml} - z_l^*)^2 \lambda_l^2 \\
s.t. \sum_{l=1}^{L}\lambda_l = 1.
\end{cases}
\tag{15}
$$

This multi-objective programming can be easily converted into a single-objective programming using the linear equal weighted summation method:

$$
\begin{cases}
\min F = \sum_{i=1}^{m}\sum_{l=1}^{L}(z_{il} - z_l^*)^2 \lambda_l^2 \\
s.t. \sum_{l=1}^{L}\lambda_l = 1.
\end{cases}
\tag{16}
$$

For the purpose of solving the quadratic programming (16), we construct a Lagrange function using a Lagrange multiplier $\eta$:

$$Lag = \sum_{i=1}^{m}\sum_{l=1}^{L}(z_{il} - z_l^*)^2\lambda_l^2 + \eta(\sum_{l=1}^{L}\lambda_l - 1) \tag{17}$$

The Hessian matrix of (17) with respect to $\lambda_l$ is a $L{\times}L$ diagonal matrix and its diagonal elements are $2\sum_{i=1}^{m}(z_{il} - z_l^*)^2 > 0$. Therefore, the Lagrange function has a minimum value, which is derived by differentiating (17) with respect to $\lambda_l$ and $\eta$ respectively:

$$\begin{cases} 2\sum_{i=1}^{m}\sum_{l=1}^{L}(z_{il} - z_l^*)^2\lambda_l + \eta = 0, \\ \sum_{l=1}^{L}\lambda_l - 1 = 0. \end{cases} \tag{18}$$

The solutions to (18) is

$$\begin{cases} \eta^* = \dfrac{1}{2\sum_{l=1}^{L}\left[\sum_{i=1}^{m}(z_{il} - z_l^*)^2\right]^{-1}}, \\ \lambda_l^* = \dfrac{1}{\sum_{l=1}^{L}\left[\sum_{i=1}^{m}(z_{il} - z_l^*)^2\right]^{-1}\sum_{i=1}^{m}(z_{il} - z_l^*)^2}. \end{cases} \tag{19}$$

Due to the fact that the constraint of (16) is a non-empty convex set, and the objective function of (16) is convex, the optimal solution (19) is the global optimal solution.

Consequently, the performance of health systems using our consensus-based model are computed as6

$$S_i = z_{il}\lambda_l^*, i = 1, 2, \ldots, m, l = 1, 2, \ldots, L. \tag{20}$$

## 3. An illustration example

In order to demonstrate and facilitate the application of the proposed consensus-based model in the process of assessing the performance of health systems, we use the simulated data about four components: effectiveness, accessibility, safety and patient-centeredness. More specifically, effectiveness refers to the degree of achieving the desired outcome, following the provision of health care services, while accessibility is the ease to reach a particular health service. Safety is a dimension that focusing the delivery of health care, which minimise risks and harm to the service users. Patient-centeredness concerns whether the healthcare services takes into account the preferences and aspirations of individual service users. The following Table 2 reports the normalized data for assessment.

On the strength of the Shannon entropy approach, the coefficient variation approach and the distance-based approach, we obtain the following three sets of relative weights associated with multiple evaluation components determined by different decision makers in the following Table 3 and Fig 2:

**Table 2. Data.**

| Health systems | Effectiveness | Accessibility | Safety | Patient-centeredness |
|---|---|---|---|---|
| 1 | 0.0484 | 0.1330 | 0.0392 | 0.1147 |
| 2 | 0.0645 | 0.0704 | 0.0576 | 0.1377 |
| 3 | 0.0860 | 0.0282 | 0.1152 | 0.0841 |
| 4 | 0.1721 | 0.0250 | 0.0392 | 0.0574 |
| 5 | 0.1033 | 0.0313 | 0.1152 | 0.0421 |
| 6 | 0.0620 | 0.0939 | 0.0576 | 0.0516 |
| 7 | 0.0553 | 0.1408 | 0.0461 | 0.0535 |
| 8 | 0.0860 | 0.0156 | 0.1843 | 0.0956 |
| 9 | 0.0860 | 0.0391 | 0.0922 | 0.0956 |
| 10 | 0.0553 | 0.1565 | 0.0461 | 0.0956 |
| 11 | 0.0704 | 0.1565 | 0.0922 | 0.0574 |
| 12 | 0.1106 | 0.1095 | 0.1152 | 0.1147 |

It is observed that different decision makers have different preferences among the assessment component. Specifically, both decision makers using the Shannon entropy and coefficient variation approaches produce the same component preference: Accessibility≻Safety≻Effectiveness≻Patient-centeredness, while the decision maker using the distance-based approach generates: Effectiveness≻Safety≻Patient-centeredness≻Accessibility. Even under the same preference relation, the weights magnitude from different decision makers are completely different. In addition, the preference of Safety is sufficiently robust across different decision makers.

Using the above weighting schemes, different assessment results determined by different decision makers are summarized as Table 4 below.

It is clearly that different decision makers produce completely different ranking of health systems. Both decision makers $cv$ and $d$ evaluate health system 12 as the best choice, while the decision maker $e$ regards health system 11 as the best. This reveals the observations that considering different weighting schemes have a significant impact on the evaluation of health systems, which necessitates developing a consensus-based decision making method to minimize the discrepancies among different decision makers.

We utilize the proposed consensus-based decision making procedure to achieve a group consensus, and derive the common weights associated with different decision makers as follows

$$(\lambda_e, \lambda_{cv}, \lambda_d) = (0.2800, 0.3563, 0.3637). \tag{21}$$

It is noticed that the opinions from decision makers $d$ and $e$ receive the most and least attention, respectively, while the opinions of decision makers $cv$ and $d$ are almost the same important.

**Table 3. Criteria weights determined by three different approaches.**

| Criteria | Shannon entropy approach | | Coefficient variation approach | | Distance-based approach | |
|---|---|---|---|---|---|---|
| | $\varphi_j$ | $w_j^e$ | $\delta_j$ | $w_j^{cv}$ | $\psi_j$ | $w_j^d$ |
| Effectiveness | 0.0275 | 0.1462 | 0.4096 | 0.2084 | 0.6642 | 0.2805 |
| Accessibility | 0.0869 | 0.4623 | 0.6587 | 0.3352 | 0.5125 | 0.2164 |
| Safety | 0.0484 | 0.2575 | 0.5291 | 0.2693 | 0.6418 | 0.2710 |
| Patient-centeredness | 0.0252 | 0.1340 | 0.3675 | 0.1870 | 0.5497 | 0.2321 |

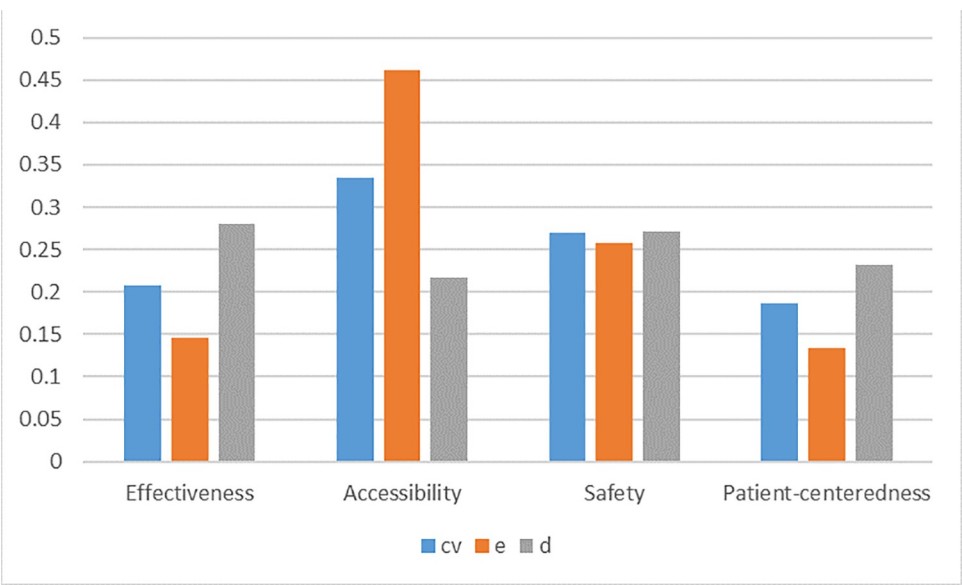

**Fig 2. Criteria weights.**

Consequently, the consensus-based decision is obtained using the proposed model, and then compared with the individual results reported in Table 4. The comparisons are demonstrated in the following Table 5 and Fig 3.

Compared with the opinions from the decision makers $e$, $cv$ and $d$, our consensus-based decision making model generates some consistent ranking positions. Specifically, the consensus-based decision model produces the same ranking as the decision maker $cv$. In addition, the rankings of HS4, HS9 and HS10 are same between decision maker $e$ and the consensus-based decision model, and the rankings of HS5 and HS12 are same between decision maker $d$ and the consensus-based decision model. HS12 is ranked at the first position by decision maker $cv$, $d$ and the consensus-based decision making model, and HS4 is ranked at the first position by decision maker $cv$, $e$ and the consensus-based decision making model.

**Table 4. Results summary.**

| Health systems | Decision Maker $cv$ | | Decision Maker $e$ | | Decision Maker $d$ | |
|---|---|---|---|---|---|---|
| | Score | Ranking | Score | Ranking | Score | Ranking |
| HS1 | 0.0867 | 5 | 0.0940 | 4 | 0.0796 | 8 |
| HS2 | 0.0783 | 7 | 0.0753 | 7 | 0.0809 | 6 |
| HS3 | 0.0741 | 8 | 0.0665 | 10 | 0.0810 | 5 |
| HS4 | 0.0655 | 12 | 0.0545 | 12 | 0.0776 | 9 |
| HS5 | 0.0709 | 10 | 0.0649 | 11 | 0.0767 | 10 |
| HS6 | 0.0696 | 11 | 0.0742 | 8 | 0.0653 | 12 |
| HS7 | 0.0811 | 6 | 0.0922 | 5 | 0.0709 | 11 |
| HS8 | 0.0907 | 4 | 0.0801 | 6 | 0.0996 | 2 |
| HS9 | 0.0737 | 9 | 0.0672 | 9 | 0.0798 | 7 |
| HS10 | 0.0943 | 3 | 0.1051 | 3 | 0.0841 | 4 |
| HS11 | 0.1027 | 2 | 0.1141 | 1 | 0.0919 | 3 |
| HS12 | 0.1122 | 1 | 0.1118 | 2 | 0.1126 | 1 |

**Table 5. Ranking comparison.**

| Health systems | cv | e | d | Our Result |
|---|---|---|---|---|
| HS1 | 5 | 4 | 8 | 5 |
| HS2 | 7 | 7 | 6 | 7 |
| HS3 | 8 | 10 | 5 | 8 |
| HS4 | 12 | 12 | 9 | 12 |
| HS5 | 10 | 11 | 10 | 10 |
| HS6 | 11 | 8 | 12 | 11 |
| HS7 | 6 | 5 | 11 | 6 |
| HS8 | 4 | 6 | 2 | 4 |
| HS9 | 9 | 9 | 7 | 9 |
| HS10 | 3 | 3 | 4 | 3 |
| HS11 | 2 | 1 | 3 | 2 |
| HS12 | 1 | 2 | 1 | 1 |

For the purpose of validating the proposed consensus-based decision model to assess health systems, the Spearman's rank correlation coefficient is computed and discussed. In statistics, Spearman's rank correlation coefficient is a nonparametric measure between the rankings of two variables, and assesses how well the relationship between two variables can be described using a monotonic function. In addition, the Spearman's rank correlation coefficient can reflect the Spearman's rank correlation coefficient. The more discordant the rankings of two variables, the smaller the Spearman's rank correlation coefficient. The formula of calculating the Spearman's rank correlation coefficient is

$$\rho_s = 1 - \frac{6 \sum_{i=1}^{m} (d_i)^2}{m^3 - m},$$ (22)

In which $d_i$ is the difference between the two ranks of each health system, and $m$ is the number of health systems. The Spearman's rank correlation coefficients between $e$, $cv$, $d$ and the consensus-based decision model are 1, 0.9231, 0.7762, respectively. This implies that the proposed model can effectively combines the results of different approaches, and thus generates a compromise and objective decision.

In summary, the proposed consensus-based decision model can effectively provide objective decision results based on the original data, without any subjective involvement of any decision makers. This implies that our model can be used as an alternative solution to the complex decision problems, especially for problem under the consideration of multiple stakeholders. Therefore, the management committee should be appropriately organized, in terms of collecting sufficient objective opinions and reaching the consensus in a reasonable manner.

## 4. Concluding remarks and future research directions

In this study, we formulate a multi-person decision making framework for assessing health systems performance. After reviewing the previous studies about this topic, we choose eeffectiveness, accessibility, safety and patient-centeredness as four components. Different stakeholders in healthcare are identified by different decision making approaches, namely, the coefficient variation approach, the Shannon entropy approach and the distance-based approach. In order to alleviate the decision discrepancy, we develop a consensus-based model to reach a group consensus about the assessment.

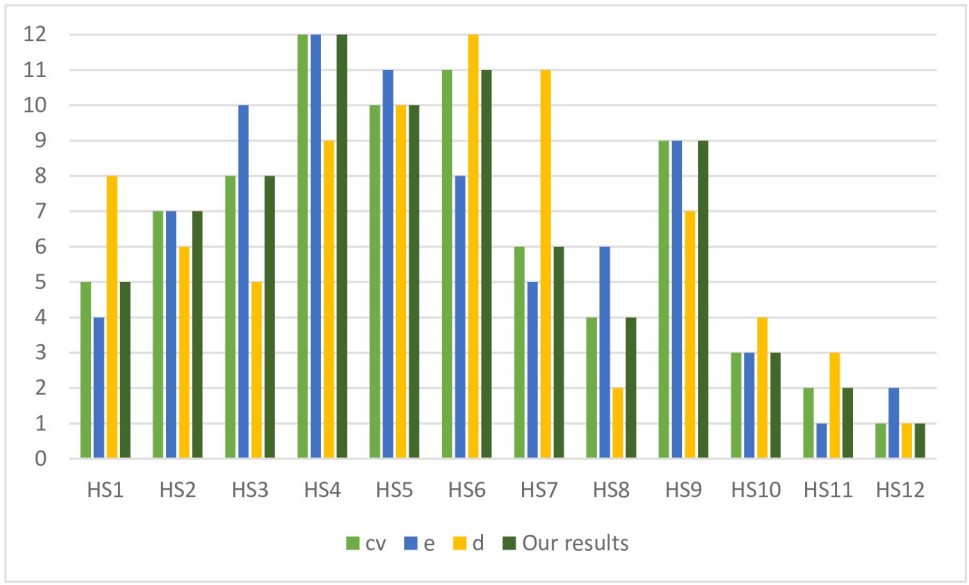

**Fig 3. Ranking comparison.**

The limitations of this study can serve as the basis for future research. First, three objective criteria weight determination approaches are employed to represent individual decision makers in this paper. The results and implications may be sensitive to the choice of criteria elicitation approaches. More approaches, such as CRITIC method, could be applied to investigate the results in future research. Second, the consensus-based model is proposed based on the ideal-point concept. The final results may be different when other consensus-achieving models are developed. Therefore, it is meaningful to compare the results from different consensus models and investigate the robustness. Third, only four components about health system performance assessment are considered in this work. Future research should have a deeper investigation in the practice and modify the assessment framework with more practical implications.

## Supporting information

**S1 Data.**
(XLSX)

## Author Contributions

**Methodology:** Yang Xu.

**Supervision:** Kin Keung Lai, W. K. J. Leung.

**Writing – original draft:** Yang Xu.

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
