## [Decision Letter · Decision Letter 0]

24 Jul 2020

PONE-D-20-20783

A Consensus-based Decision Model for Assessing the Health Systems

PLOS ONE

Dear Dr. Lai,

Thank you for submitting your manuscript to PLOS ONE. After careful consideration, we feel that it has merit but does not fully meet PLOS ONE’s publication criteria as it currently stands. Therefore, we invite you to submit a revised version of the manuscript that addresses the points raised during the review process.

We look forward to receiving your revised manuscript.

Kind regards,

Dragan Pamucar

Academic Editor

PLOS ONE

Journal Requirements:

2. Thank you very much for your submission to PLOS ONE. Before we proceed, we kindly ask that you address the following:

*Please explain the rationale for the development of your method in light of recent research in this area, clearly indicating which problem with existing method you are addressing.

*Please clearly report at the beginning of your methods or results section which the key performance measures were to establish validity and utility of your method. Please also report clearly which statistical analysis was used to establish robustness of performance measures.

*Please note that PLOS ONE requires that experiments, statistics, and other analyses must be performed to a high technical standard and described in sufficient detail to allow for reproducibility of the study (http://journals.plos.org/plosone/s/criteria-for-publication#loc-3). To demonstrate the performance of the method, we would expect comparisons to be drawn between existing state-of-the-art methods.

Thank you for your attention to these queries.

3. Thank you for including your funding statement; "The funders had no role in study design, data collection and analysis, decision to publish, or preparation of the manuscript."

Reviewers' comments:

Reviewer's Responses to Questions

**Comments to the Author**

1. Is the manuscript technically sound, and do the data support the conclusions?

Reviewer #1: Yes

Reviewer #2: Yes

2. Has the statistical analysis been performed appropriately and rigorously? 

Reviewer #1: Yes

Reviewer #2: Yes

3. Have the authors made all data underlying the findings in their manuscript fully available?

Reviewer #1: Yes

Reviewer #2: Yes

4. Is the manuscript presented in an intelligible fashion and written in standard English?

Reviewer #1: Yes

Reviewer #2: Yes

5. Review Comments to the Author

Reviewer #1: Dear editor,

Thank you for sending me for review the paper “A Consensus-based Decision Model for Assessing the Health Systems”. This paper intends to investigate Entropy method for determining criteria weights and TOPSIS evaluation method group decision making. The authors implemented hybrid MCDM framework for evaluation in healthcare field.

Presented methodology has a great potential in decision making process and I am giving a support to the authors for investigation this topic. The strengths of this paper are: Relevant topic; Flow of the paper; and Explanation of the methods. However, the author(s) need to consider the following points as limitation or further scope for refining the paper:

- Introduction should be clearly stated research questions and targets first. Then answer several questions: Why is the topic important (or why do you study on it)? What are the research questions? What are your contributions? Why is to propose this particular methods? The last two questions are answered in some parts in the Introduction section. But, the answer is not presented in a proper way. You should provide more information in this regard.

- Need to highlight the novelty of study in the introduction.

- I suggest authors to clearly summarize what specific advantages brings your approach. Enrich your Introduction section with more explanation: Why do you present this approach? Why you use Entropy method for criteria weighting and not the other objective methods like CRITIC or FANMA method?

- Why did you use objective methodology for determining criteria weights and not subjective methodologies like AHP, BWM, LBWA etc? I must stress that there are numerous limitations of Entropy method that authors should have on their minds. For example, if only one value in decision matrix is above/below the other values for 20-30% (within the same criteria) leads drastically to increasing criteria weight of that criteria. That limitation is presented in almost all objective methods. In my opinion this facts should not be neglected. So, why we need objective weights?

- Remove lumped references. All references cited in the text should be explained and discussed in the text. Remove some old references published before 2017-2018. Also, literature review should be presented in a better way. You should discuss application of various MCDM tools different fields, especially in healthcare field. You should update your literature review with a papers published in last two-three years, and remove old references. I suggest authors to read and cite below interesting references: Roy, J., Adhikary, K., Kar, S., & Pamucar, D. (2018). A rough strength relational DEMATEL model for analysing the key success factors of hospital service quality. Decision Making: Applications in Management and Engineering, 1(1), 121-142.;

Badi, I., Abdulshahed, A., Shetwan, A., & Eltayeb, W. (2019). Evaluation of solid waste treatment methods in Libya by using the analytic hierarchy process. Decision Making: Applications in Management and Engineering, 2(2), 19-35.

Biswas, S., Bandyopadhyay, G., Guha, B., & Bhattacharjee, M. (2019). An ensemble approach for portfolio selection in a multi-criteria decision making framework. Decision Making: Applications in Management and Engineering, 2(2), 138-158.

- Add flowchart of proposed methodology and follow that flowchart steps in case study.

- Case study should be better organized. The calculations should be deeply presented and follow the methodology presented in methodology section. Add more deep calculations in case study section.

- Add sensitivity analysis and validation of the results.

- The problem on which this present method is applied has significant social and managerial implications. How the method can address those implications need to be included.

- Conclusion- Add future scope. Also, how the proposed method can be applicable to other real life problems need to be mentioned. Add limitations of proposed model. Do not use bullets or numerations in this section.

I will review revised paper with my pleasure.

Reviewer #2: The paper analyses an actual topic and it is interesting to potential readers. The paper is well prepared, it contains all required parts of a scientific paper.

I have only one critical comment. References are up to 2016. Therefore the authors should update the literature review.

6. PLOS authors have the option to publish the peer review history of their article (what does this mean?). If published, this will include your full peer review and any attached files.

Reviewer #1: No

Reviewer #2: No

---

## [Author Response · Author response to Decision Letter 0]

31 Jul 2020

Many thanks for your valuable comments on our manuscript. We accordingly make the revisions and mark them using RED in the paper.

Reviewer #1: Dear editor, Thank you for sending me for review the paper “A Consensus-based Decision Model for Assessing the Health Systems”. This paper intends to investigate Entropy method for determining criteria weights and TOPSIS evaluation method group decision making. The authors implemented hybrid MCDM framework for evaluation in healthcare field.

Presented methodology has a great potential in decision making process and I am giving a support to the authors for investigation this topic. The strengths of this paper are: Relevant topic; Flow of the paper; and Explanation of the methods. However, the author(s) need to consider the following points as limitation or further scope for refining the paper:

- Introduction should be clearly stated research questions and targets first. Then answer several questions: Why is the topic important (or why do you study on it)? What are the research questions? What are your contributions? Why is to propose this particular methods? The last two questions are answered in some parts in the Introduction section. But, the answer is not presented in a proper way. You should provide more information in this regard.

Many thanks for your comments. (1) We discuss the importance of this study in paragraph 1. (2) The research questions are described in Page 5: (1) how to define the individual stakeholders of health systems? (2) how to achieve the consensus among different stakeholders? (3) the contributions and novelties are summarized in Pages 5&6. (4) The present paper is motivated by the observation that in the process of performance assessment, not only the preferences associate with evaluation criteria may exhibit a substantial degree of variability, but also different members of the decision committee have different opinions, which are extremely difficult to achieve a group consensus (Csaszar and Eggers, 2013; Melkonyan and Safra, 2016). In this sense, this work proposes a consensus-based decision model to assess the health systems. 

- Need to highlight the novelty of study in the introduction.

Many thanks for your comments. The contributions and novelties of this study are summarized in Section 1.

- I suggest authors to clearly summarize what specific advantages brings your approach. Enrich your Introduction section with more explanation: Why do you present this approach? Why you use Entropy method for criteria weighting and not the other objective methods like CRITIC or FANMA method?

Many thanks for your comments. There exist many other objective weight elicitation approaches to assessing the health systems. We choose the three approaches for illustrating the working procedure of the proposed consensus-based decision model. The other objective methods would be investigated in future research. 

- Why did you use objective methodology for determining criteria weights and not subjective methodologies like AHP, BWM, LBWA etc? I must stress that there are numerous limitations of Entropy method that authors should have on their minds. For example, if only one value in decision matrix is above/below the other values for 20-30% (within the same criteria) leads drastically to increasing criteria weight of that criteria. That limitation is presented in almost all objective methods. In my opinion this facts should not be neglected. So, why we need objective weights?

Many thanks for your comments. The main advantage of these objective approaches is the reduction of decision bias in terms of ignoring the subjective judgments of the individual stakeholders. Objective criteria weight determination approaches are usually applicable when individual stakeholders disagree on the exact values of criteria weights (Yu & Lai, 2011). Speciﬁcally, the rationale behind objective criteria weight determination approaches is that the importance degree of a criterion is a function of the information conveyed by this criterion, relative to a whole set of alternatives.

- Remove lumped references. All references cited in the text should be explained and discussed in the text. Remove some old references published before 2017-2018. Also, literature review should be presented in a better way. You should discuss application of various MCDM tools different fields, especially in healthcare field. You should update your literature review with papers published in last two-three years, and remove old references. I suggest authors to read and cite below interesting references: Roy, J., Adhikary, K., Kar, S., & Pamucar, D. (2018). A rough strength relational DEMATEL model for analysing the key success factors of hospital service quality. Decision Making: Applications in Management and Engineering, 1(1), 121-142.;

Badi, I., Abdulshahed, A., Shetwan, A., & Eltayeb, W. (2019). Evaluation of solid waste treatment methods in Libya by using the analytic hierarchy process. Decision Making: Applications in Management and Engineering, 2(2), 19-35.

Biswas, S., Bandyopadhyay, G., Guha, B., & Bhattacharjee, M. (2019). An ensemble approach for portfolio selection in a multi-criteria decision making framework. Decision Making: Applications in Management and Engineering, 2(2), 138-158.

Many thanks for your comments. The literature part has been updated according to your comments and some other new publications. 

- Add flowchart of proposed methodology and follow that flowchart steps in case study.

Many thanks for your comments. The flowchart is added to Section 2.

- Case study should be better organized. The calculations should be deeply presented and follow the methodology presented in methodology section. Add more deep calculations in case study section.

Many thanks for your comments. We further elaborate the computation process of the proposed methodology in case study section (Section 3). 

- Add sensitivity analysis and validation of the results.

Many thanks for your comments. For the purpose of validating the proposed consensus-based decision model to assess health systems, the Spearman’s rank correlation coefficient is computed and discussed. Please refer to Section 3. 

- The problem on which this present method is applied has significant social and managerial implications. How the method can address those implications need to be included.

Many thanks for your comments. The implications are added in the last paragraph of Section 3. 

- Conclusion- Add future scope. Also, how the proposed method can be applicable to other real life problems need to be mentioned. Add limitations of proposed model. Do not use bullets or numerations in this section.

Many thanks for your comments. Limitations and future scope are added to Section 4. 

Reviewer #2: The paper analyses an actual topic and it is interesting to potential readers. The paper is well prepared, it contains all required parts of a scientific paper.

I have only one critical comment. References are up to 2016. Therefore the authors should update the literature review.

Many thanks for your valuable comments. We have added several references after 2016 in Section 1.

---

## [Decision Letter · Decision Letter 1]

5 Aug 2020

A Consensus-based Decision Model for Assessing the Health Systems

PONE-D-20-20783R1

Dear Dr. Lai,

We’re pleased to inform you that your manuscript has been judged scientifically suitable for publication and will be formally accepted for publication once it meets all outstanding technical requirements.

Kind regards,

Dragan Pamucar

Academic Editor

PLOS ONE

Additional Editor Comments (optional):

Reviewers' comments:

Reviewer's Responses to Questions

**Comments to the Author**

1. If the authors have adequately addressed your comments raised in a previous round of review and you feel that this manuscript is now acceptable for publication, you may indicate that here to bypass the “Comments to the Author” section, enter your conflict of interest statement in the “Confidential to Editor” section, and submit your "Accept" recommendation.

Reviewer #1: All comments have been addressed

Reviewer #2: All comments have been addressed

2. Is the manuscript technically sound, and do the data support the conclusions?

Reviewer #1: Yes

Reviewer #2: Yes

3. Has the statistical analysis been performed appropriately and rigorously? 

Reviewer #1: Yes

Reviewer #2: Yes

4. Have the authors made all data underlying the findings in their manuscript fully available?

Reviewer #1: Yes

Reviewer #2: Yes

5. Is the manuscript presented in an intelligible fashion and written in standard English?

Reviewer #1: Yes

Reviewer #2: Yes

6. Review Comments to the Author

Reviewer #1: All the reviewers' comments have been addressed carefully and sufficiently, the revisions are rational from my point of view, I think the current version of the paper can be accepted.

Reviewer #2: Several new references are added, but there could be more of them. Nvertheless, the paper can be accepted.

7. PLOS authors have the option to publish the peer review history of their article (what does this mean?). If published, this will include your full peer review and any attached files.

Reviewer #1: No

Reviewer #2: No

---

## [Editor Report · Acceptance letter]

7 Aug 2020

PONE-D-20-20783R1 

A Consensus-based Decision Model for Assessing the Health Systems 

Dear Dr. Lai:

I'm pleased to inform you that your manuscript has been deemed suitable for publication in PLOS ONE. Congratulations! Your manuscript is now with our production department. 

Kind regards, 

on behalf of

Dr. Dragan Pamucar 

Academic Editor

PLOS ONE